# Causes of model dry and warm bias over central U.S. and impact on climate projections

Yanluan Lin [1], Wenhao Dong [1], Minghua Zhang[2,3], Yuanyu Xie[1], Wei Xue[1,4], Jianbin Huang[1] & Yong Luo[1]

Climate models show a conspicuous summer warm and dry bias over the central United States. Using results from 19 climate models in the Coupled Model Intercomparison Project Phase 5 (CMIP5), we report a persistent dependence of warm bias on dry bias with the precipitation deficit leading the warm bias over this region. The precipitation deficit is associated with the widespread failure of models in capturing strong rainfall events in summer over the central U.S. A robust linear relationship between the projected warming and the present-day warm bias enables us to empirically correct future temperature projections. By the end of the 21st century under the RCP8.5 scenario, the corrections substantially narrow the intermodel spread of the projections and reduce the projected temperature by 2.5 K, resulting mainly from the removal of the warm bias. Instead of a sharp decrease, after this correction the projected precipitation is nearly neutral for all scenarios.

[1] Ministry of Education Key Laboratory for Earth System Modeling, Department of Earth System Science, and Joint Center for Global Change Studies (JCGCS), Tsinghua University, Beijing 100084, China. [2] School of Marine and Atmospheric Sciences, Stony Brook University, Stony Brook, NY 11794-5000, USA. [3] Institute of Atmospheric Physics, Chinese Academy of Sciences, Beijing 100029, China. [4] Department of C omputer Science and Technology, Tsinghua University, Beijing 100084, China. Yanluan Lin and Wenhao Dong contributed equally to this work. Correspondence and requests for materials should be addressed to Y.L. (email: yanluan@tsinghua.edu.cn) or to M.Z. (email: minghua.zhang@stonybrook.edu)

Well-planned adaptation and mitigation policies require reliable information on how future climate changes. However, despite decades of efforts to improve model performances, systematic biases still exist over highly populated regions on continental scales in the majority of climate models[1–6]. Compelling evidences have been presented that future climate change is obscured by these modeling uncertainties[2, 6] which is likely to remain so for years to come when results of climate models are directly used. Identifying the origins of these systematic biases and determining their potential impacts on future climate changes are thus immensely important and crucial for adaptation guidance and policy making. This topic is currently the subject of intense research in the climate community. For example, in the most recent round of Coupled Model Intercomparison Project Phase 6 (CMIP6), understanding and reducing model systematic biases is one of the three major tasks of the community effort[7].

Among the model biases, one of the most conspicuous is the warm and dry bias over the central U.S. (CUS) in the summer (June–August) that has persisted in many generations of regional and global climate models[1, 3, 4]. It is believed that the land (soil moisture)–atmosphere feedback is crucial in causing the model deficiencies[1, 8, 9]. In the water-limited CUS region[10, 11], soil moisture deficit can directly impact the partitioning of diurnal radiative energy between latent and sensible heat fluxes through local evapotranspiration[4]. Consequently, it influences the formation of shallow cumulus through planetary boundary layer interaction[3]. Soil moisture deficit and the resultant underestimated shallow cumulus can lead to an excess of surface temperature and, meanwhile, reduce the formation of precipitation. Though these feedbacks have been extensively discussed, the origins of the model biases are still unclear. More importantly, a warming and drying future is projected over the CUS by current models[12], which has been widely recognized to have significant socioeconomic and agriculture impacts[13]. Given the large warm and dry bias over this region, our confidence in the model projections is overshadowed by potential uncertainties. How the warm and dry bias impact future climate projections remains unknown and needs to be evaluated. Here we identify that the warm and dry bias over the CUS in GCMs is likely triggered by the precipitation deficit arising from the model's failure in capturing the large precipitation events, followed by land–atmosphere feedbacks. We make use of the robust linear relationship between the projected warming and the present-day warm bias to correct the future projections of temperature. The future projected precipitation is empirically corrected on the basis of the dependence of warm bias on dry bias. Results show that current GCMs are likely overestimating future warming and drying over the CUS.

## Results

**Correspondence of warm bias and dry bias.** The CUS surface climate biases in models have been documented in several previous studies[1]. In the historical simulations from 19 models participating in Phase 5 of the Coupled Model Intercomparison Project (CMIP5)[14], the multi-model mean (MMM) biases of summertime temperature and precipitation over the CUS during the reference period 1979–2005 are up to 3° warmer (Fig. 1a), in which 14 out of 19 of the models agree on the sign of the warm bias, and up to 43% drier (Fig. 1b), in which 17 out of 19 of the

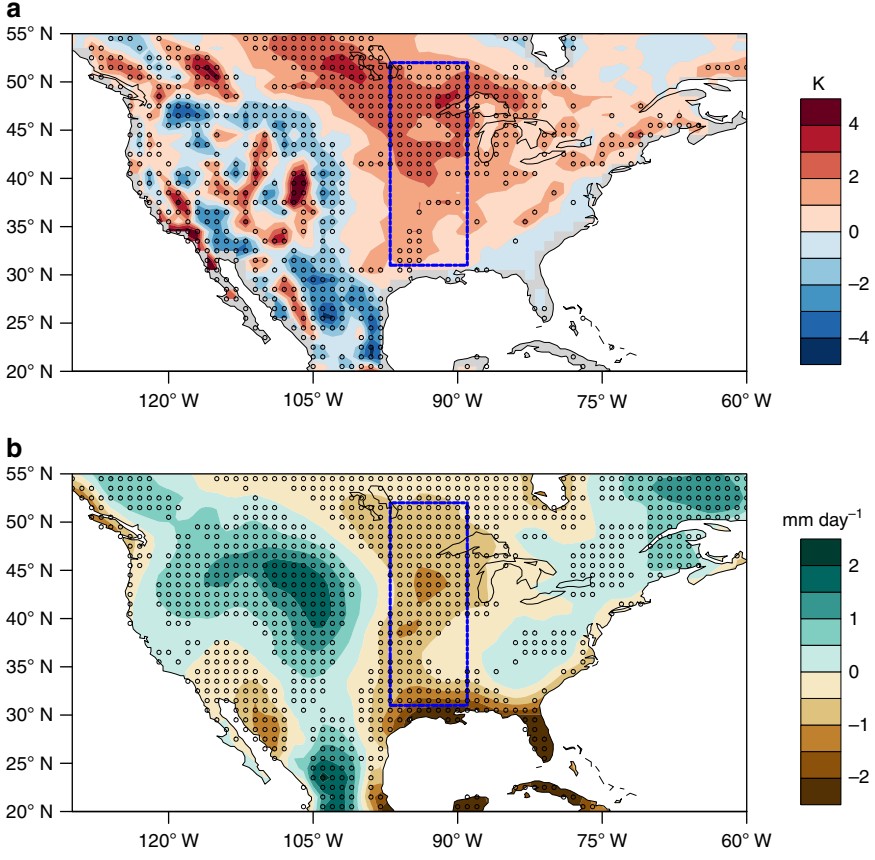

**Fig. 1** Geographical biases of temperature and precipitation in CMIP5 models. **a** Multi-model mean of temperature biases and **b** precipitation biases in summer during 1979–2005 from 19 CMIP5 historical simulations. Regions where at least two thirds of the models (i.e., 13 out of 19 CMIP5 models) agree on the sign of the difference are marked with black circles. The blue rectangle (31–52° N, 262–271° E) indicates the central U.S.

models agree on the sign of the dry bias. The MMM warm and dry biases are present throughout the year, but they peak in the summer (Supplementary Table 1).

A scatter plot of temperature bias and precipitation bias clearly demonstrates a close connection between the two among the models (Fig. 2a), which can be well fitted by a linear function:

$$T_{bias} = 3.28(-P_{bias}) - 0.66.$$

The two biases are strongly related ($r^2 = 0.65$; $P < 1 \times 10^{-4}$), with larger dry bias corresponding to larger warm bias. Such a significant correlation indicates that the model biases and the intermodel differences may share the same underlying physical cause. The MMM area-averaged warm bias is $1.8 \pm 0.9$ ($\pm$ one s. d.; K) while the MMM dry bias is $-0.8 \pm 0.4$ (mm d$^{-1}$) (or $-23 \pm 11\%$) for the reference period from 1979 to 2005. Individual model performance differs greatly with the ranges of temperature and precipitation biases spreading over 8 K and 2 mm d$^{-1}$, respectively. For instance, one model simulates an area-averaged warm bias up to 6.6 K and one model simulates an area-averaged dry bias up to $-1.8$ mm d$^{-1}$ or 51% (Supplementary Table 1).

**Precipitation deficit leads warm bias.** Examination of the seasonal evolution of the temperature and precipitation biases in different CMIP5 models indicates precipitation bias preceding that of temperature. This can be illustrated by selecting two groups of models—the good and bad, based on their performances in historical simulations. The good group includes models with warm and dry bias much less than the MMM bias (on average, $-0.3$ K for temperature and $-0.1$ mm d$^{-1}$ for precipitation), whereas the bad group includes models with biases larger than the MMM biases (on average, 4.8 K and $-1.4$ mm d$^{-1}$, approximately two times larger than the MMM biases). The

classification is indicated by the circles in Fig. 2a, and each group has five models. In the good model composite, the peaks of precipitation deficit and the warm bias occur almost simultaneously from July to August (Fig. 2b). By contrast, in the bad group, the peak of precipitation deficit leads the warm bias by about 1 month. The precipitation bias reaches its maximum during June to July, while the temperature bias peaks from July to August (Fig. 2c).

This time lag is consistent with the cooling effects of precipitation on temperatures, which can be illustrated by using observations of surface temperature and precipitation at the SGP site of the Atmospheric Radiation Measurement (ARM)[15] Program. We categorize the daily rainfall into three intensities and examined the associated changes of surface temperature in the subsequent 5 days (see "Methods"). Following a heavy precipitation event, temperature drops quickly and substantially and then recovers gradually (Supplementary Fig. 1a). The stronger the rainfall, the larger the magnitude of the temperature decrease, and the longer duration of the cooling effect. This suggests that surface temperature following a rain event will be misrepresented if the model could not simulate the precipitation accurately, especially for strong rainfall events. Consequently, the warm bias can stem from a precipitation deficit. After the precipitation events, temperature ultimately recovers, primarily related to the increased net downward radiation (Supplementary Fig. 1b) associated with reduced clouds.

But what is the origin of the precipitation deficit in the models? Summertime precipitation over the CUS, characterized by a dominant nocturnal peak, is mostly produced by mesoscale convective systems[1, 16–18]. These systems are typically triggered over the Rocky Mountains on the preceding afternoon[18, 19] and they propagate eastward to tap the moisture source transported from the Gulf of Mexico by the nocturnal lower-level jet[20]. Some of them organize into squall lines. Notoriously, current GCMs cannot accurately simulate these convective systems due to the

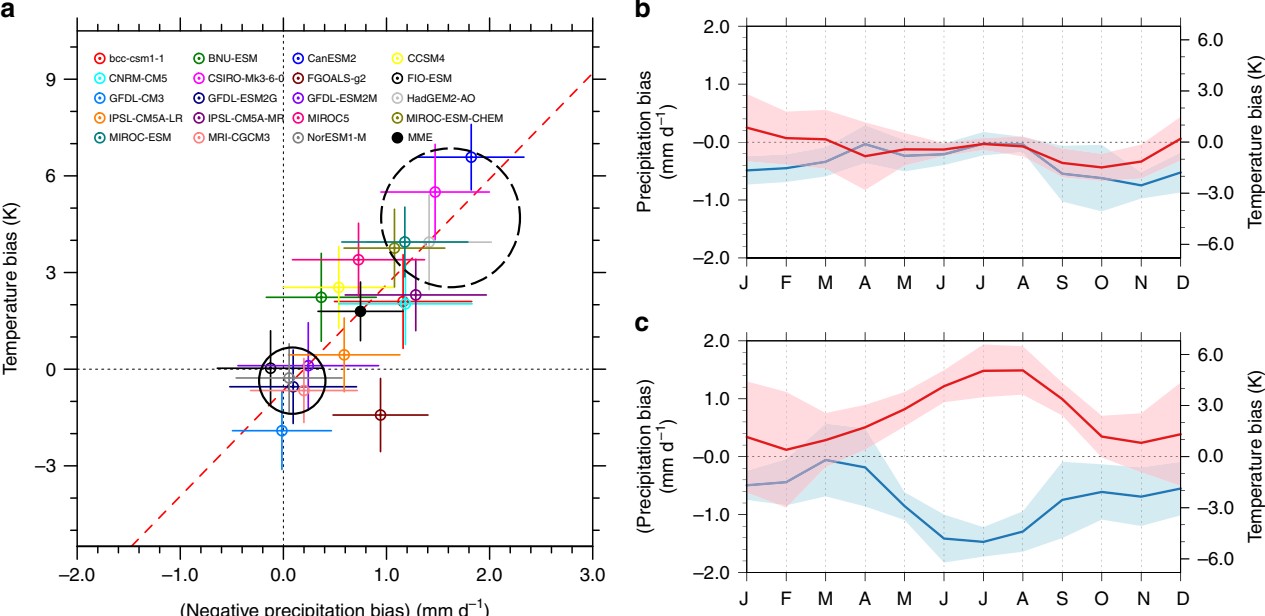

**Fig. 2** Characteristics of temperature bias and precipitation bias. **a** Scatter plot of the summertime temperature bias and precipitation bias over the central U.S. from 19 CMIP5 historical simulations during the reference period 1979–2005. Solid dot indicates the multi-model mean while horizontal and vertical colored lines indicate one standard deviation The red dash line indicates the least-square linear fit among the models. The solid (dash) circle contains the models selected as good (bad) group. **b** Seasonal evolution of temperature and precipitation biases for the good group. **c** Same as **b** but for the bad group. The blue lines in **b**, **c** use the left y-axes, indicating precipitation, while the red lines use the right y-axes, indicating temperature. Shaded regions in **b**, **c** indicate one standard deviation

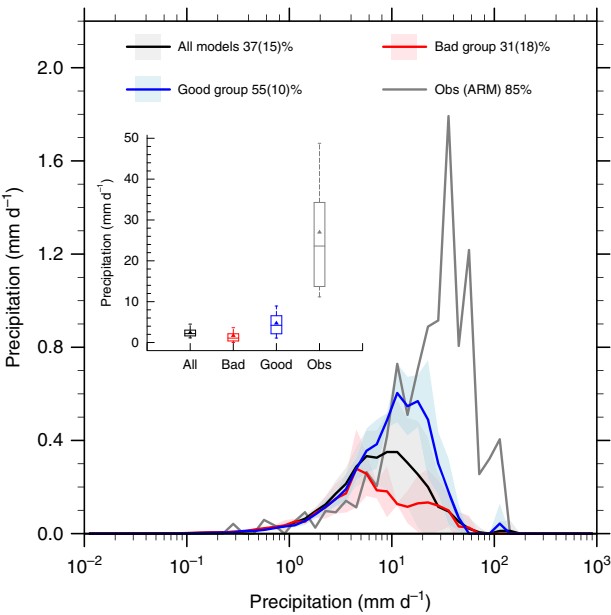

**Fig. 3** Daily precipitation distribution. Distribution of daily precipitation for all models, good group, bad group, as well as that observed by Atmospheric Radiation Measurement (ARM) Program during 1996–2005 at the SGP site. The inset plot is the boxplot of daily precipitation. The boxes indicate the 10%-quantile, 25%-quantile, median, 75%-quantile, and 90%-quantile, respectively. Solid triangles stand for the mean values

fact that the convective parameterizations in the current generation of climate models have not been intended to work well for these systems[21]. As a result, these models are largely incapable of simulating the strong precipitation events in the CUS, not to mention the right nocturnal timing[17, 19]. This is evident in Fig. 3, in which we compare the intensity frequency distribution of model daily precipitation at the grid point closest to the SGP site during 1996–2005 with ARM observations. Note that the ARM precipitation has been area-averaged with a similar spatial coverage as a model grid box. The models severely underestimate the rainfall intensity and the occurrence frequency of heavy rainfall events (here defined as daily precipitation ≥10 mm). Heavy rainfall events contribute to 85% of the total precipitation in ARM observation, while the percentage simulated by the models is only 37% ± 15%. Consistent with the previously discussed results of the good and bad models, the good group has relatively larger probability of heavy rainfall occurrence (55% ± 10%) than the bad group (31% ± 18%).

Besides the precipitation bias, other deficiencies reported in the model over this area, such as underestimated shallow cumulus[22], misrepresentation of land soil properties and land–atmosphere processes[23], and possibly misrepresented anvil clouds[24], could also contribute to the warm bias. Once an initial dry bias is formed, the buildup of the warm bias can occur through a series of land–atmosphere feedback processes on different time scales: a deficit in precipitation not only alters evaporative cooling due to evapotranspiration[1, 4], but also indirectly reduces attendant cloud formation and infiltrated soil moisture[3, 9, 10, 25], leading to increased solar radiation[4], and thus the amplification of surface warming during the following days. The temperature, from an energetic perspective, is predominantly constrained by the radiant and turbulent fluxes. These fluxes are influenced by several environmental factors on different time scales[26], including soil moisture, relative humidity, and cloud cover that could be strongly impacted by the preceding nocturnal precipitation. To quantify the impacts of different rainfall associated processes on

temperature, we utilize the comprehensive variables from ARM best estimate (ARMBE) data archive for detailed analysis. We first compare the diurnal changes in the surface temperature, cloud fraction, and surface energy budget terms between precipitating days and non-precipitating days (Table 1). As shown in the diurnal evolution (Supplementary Fig. 2), the temperature contrast between the different composites peaks during daytime with the value to be 4.8 K (3.4 K if we take the daily average). This indicates that although the reduced temperature in rainy days may partly result from the direct rain evaporative cooling including cold pool associated with the MCS passage, it mainly results from the processes during the daytime. All the daytime radiant and turbulent fluxes are smaller during the precipitating days than those during the non-precipitating days. Clouds play a crucial role in regulating the energy budget given its strong interaction with both solar and infrared radiation. Shallow cumulus, usually occurring during daytime with the growth of the boundary layer and a gradually lifted cloud base under 3 km, has a typical fraction of 10–12% over this area. It strongly impacts the shortwave radiation. The cloud fraction of shallow cumulus is larger than 15% during precipitating days while it is about 5% during non-precipitating days (Fig. 4a, b). The total cloud fraction is about two times larger in precipitating days compared to non-precipitating days. This leads directly to a reduction in net shortwave absorption of 107.2 W m$^{-2}$ with 32.4 W m$^{-2}$ smaller net longwave emission. The net reduced radiative heating (74.8 W m$^{-2}$) is approximately balanced by the decreased latent heat (37.9 W m$^{-2}$) and sensible heat (36.1 W m$^{-2}$) fluxes (the residual ground heat flux term is usually small relative to other components). We further divide precipitating days into three categories by intensity (Supplementary Table 2). The average differences of nocturnal precipitation amounts correlate well with the correspondent daytime temperature changes with $r = -0.92$ ($P < 0.05$), indicating that the larger the nocturnal precipitation, the larger daytime temperature drop. And consistent with Phillips and Klein[11], the net shortwave radiation decreases quasi-linearly with increasing cloud fraction while a highly positive correlation between the net surface longwave fluxes with cloud cover is found. Sensible and latent heat fluxes vary inversely with cloud fraction (Supplementary Table 2). Switching the averaging periods does not change the conclusion. As a result, on a diurnal time scale, the daytime temperature drop in rainy days is predominately associated with the reduced solar radiation.

During the following days after a rainfall event, however, the inherent complexities in land–atmosphere interactions might make these processes more complicated. For instance, soil moisture is proved to have profound impacts on land–atmosphere interaction due to its long memory[8]. It could impact cloud formation in the aftermath of a precipitation event even without any rainfall in the following several days. Gradual drying of the soil after a rainfall event has been found to impact the land–atmosphere feedback at lags between 4 and 8 days[11]. To quantify impacts of rainfall on temperature in the following days, we contrast the surface energy terms between two composites: a wet precondition composite requiring five consecutive non-precipitating days following a rainfall event and a dry precondition composite requiring five consecutive non-precipitating days following prior five consecutive non-precipitating days. The prior five consecutive non-precipitating days in dry precondition composite are adopted here to minimize the influence of preceding rainfall events (see "Methods"). This classification makes the two composites independent of each other and thus enables us to quantify the rainfall impact on temperature. Compared with dry precondition, surface temperature after the rainfall is about 2.8 K lower under wet precondition, which, from the surface energy budget, is primarily associated with the

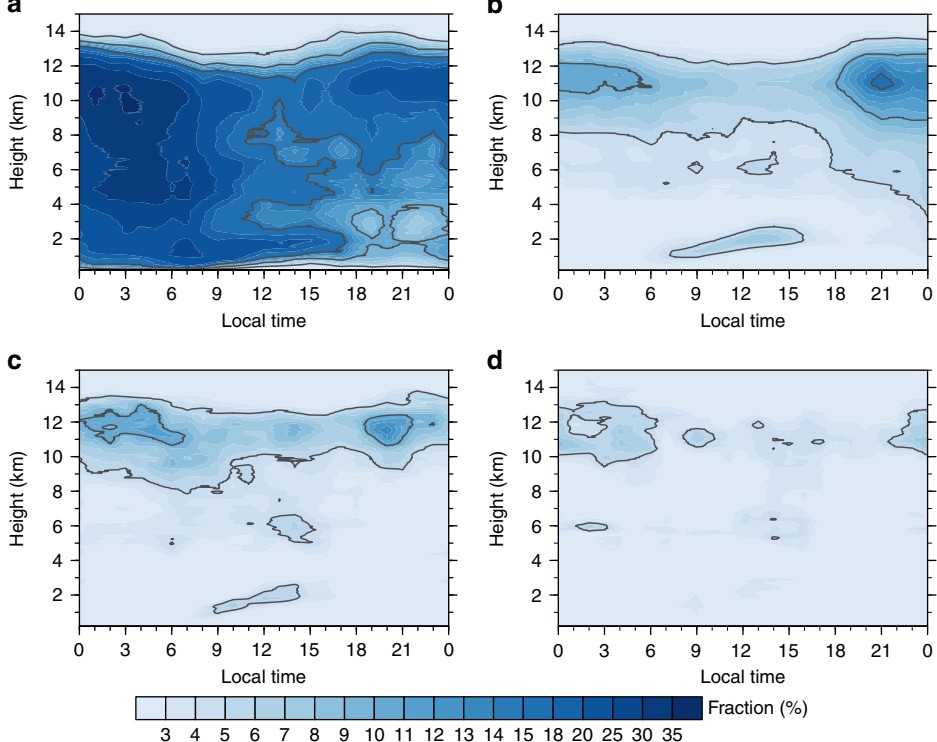

**Fig. 4** Composite diurnal variations of cloud fraction. **a** Diurnal cycle of cloud fraction averaged over precipitating days, **b** non-precipitating days, **c** five consecutive non-precipitating days under wet precondition, and **d** five consecutive non-precipitating days under dry precondition. The sample sizes are 288, 945, 49, and 73, respectively. Contours indicate 5, 10, and 15% cloud fraction

**Table 1 Contrast of precipitation, temperature, cloud fraction, sensible heat, latent heat, and radiation terms between precipitating days and non-precipitating days**

| | $P$ (mm d$^{-1}$) | $T$ (°C) | Cloud fraction (%) | | Latent heat (W m$^{-2}$) | Sensible heat (W m$^{-2}$) | SW (W m$^{-2}$) | | LW (W m$^{-2}$) | |
|---|---|---|---|---|---|---|---|---|---|---|
| | | | 3 Types | Total | | | Upward & downward | Net | Upward & downward | Net |
| Precipitating days (288) | 22.08 | 25.43 | 9.99 (low) 12.85 (mid) 39.64 (high) | 62.96 | −176.83 | −74.57 | 396.07 (downward) −78.35 (upward) | 317.72 | 405.46 (downward) −455.41 (upward) | −49.95 |
| Non-precipitating days (945) | 0 | 30.22 | 6.46 (low) 7.71 (mid) 15.14 (high) | 29.68 | −214.72 | −110.69 | 535.88 (downward) −111 (upward) | 424.88 | 406.36 (downward) −488.67 (upward) | −82.31 |
| Precipitating minus non-precipitating | 22.08 | −4.79 | 3.53 (low) 5.14 (mid) 24.50 (high) | 33.28 | 37.89 | 36.12 | −139.81 (downward) 32.65 (upward) | −107.16 | −0.9 (downward) 33.26 (upward) | 32.36 |

The precipitation is averaged between 0000 and 0600 local solar time (LST) while the other variables are averaged between 0600 and 1800 LST. The radiant and turbulent fluxes are positive downward. The numbers in the parentheses indicated the sample size
Note: The total cloud fraction is not equal to the sum of three different types of cloud fraction due to the overlap

enhanced latent heat fluxes (Supplementary Table 3). Larger cloud fraction is noted under wet precondition, particularly, shallow cumulus is more prevalent due to the strong coupling with soil moisture (Fig. 4c). In contrast, shallow cumulus is rather limited under dry precondition (Fig. 4d). This indicates that soil properties and land surface processes after the rainfall infiltration can indirectly regulate the subsequent low cloud formation. These features are also evident in the evolution of other variables in the surface energy budget (Supplementary Fig. 3). Low clouds gradually decrease to a level (~5%) similar to that in the dry composite 3 days after a rainfall event. This is likely due to the subsequent gradual evaporative drying of the soil in the absence of additional precipitation. One thing worth noting is that despite the larger cloud fraction under wet precondition, there is more downward solar radiation near the surface (Supplementary

Table 3). This lies in the seasonal variation of the solar insolation at the top of the atmosphere. It decreases progressively from June to September. And wet precondition composite has larger proportion in June (28.6%) than dry precondition composite (6.8%). In fact, the scaled surface downward shortwave radiation increases gradually to that under dry precondition, consistent with the evolution of cloud fraction (Supplementary Fig. 3e). These results imply that except the immediate temperature drop on rainy days, such an impact could last for several days in the aftermath of a precipitation event due to the land–atmosphere interaction mediated by soil moisture. Other properties, like the lower atmosphere's aerodynamic resistance and humidity saturation deficit, could complicate the strength of land–atmosphere coupling[11, 27].

The above analyses indicate that if a rainfall event, especially a large episode, is missed in the model, the associated surface

temperature decrease would be misrepresented through complex land–atmosphere interactions on different time scales. The initial dry bias acts as the source of the eventual simulated warm bias, which is amplified by other model deficiencies, such as underestimated shallow cumulus and misrepresented soil moisture processes, among others. These results provide strong evidence that one culprit of the systematic warm and dry biases in climate models is likely a combination of the inability of climate models in simulating the strong precipitation events caused by the mesoscale convective systems with other potential model deficiencies. The CUS happens to be located in a region where these convective systems occur more frequently than in other regions[16, 17] and where the surface is moisture limited that allows strong land–atmosphere feedbacks[1, 10]. The inherent limitation of current models in capturing these convective systems and the two ingredients of climate regime over the CUS—the mesoscale systems and the strong land–atmosphere coupling—can conspire to create the systematic model biases over the CUS.

**Correction for projected temperature.** How would the identified systematic biases impact future projections of climate change in CUS? We calculate the projected CUS-averaged temperature and precipitation change over the 20 years at the end of 21st century in all the three RCP scenarios relative to the 20 years at the end of 20th century in historical simulation (Supplementary Table 4). The projected temperature and precipitation change vary greatly among these models with an overall warming and drying. On average, taking RCP8.5 scenario as an example, model projected warming is nearly 6 K with a slight drying about 0.2 mm d$^{-1}$ by the end of 21st century, indicating a warmer and drier future over the CUS[12]. An important question is how these biases may affect the climate change projections. We start with the temperature bias since the model temperature changes are relatively consistent among the models. Given a model with a systematic warm bias, future warming can be assumed to be the sum of one related to the radiative forcing and the other induced by the bias itself. To estimate the impact of model systematic warm bias on future projections, we regress the projected temperature change with the identified systematic biases. We note robust linear relationships between temperature bias and its future projected change in all three RCP scenarios (Fig. 5), with linear regression coefficient to be 0.27, 0.31, 0.41 and $r^2 = 0.54$, 0.40, and 0.44 ($P < 0.01$). The linear least-square fitting lines exclude model 9, which has known limitations for too large climate sensitivity[28] (but is included here for completeness). This suggests that a model with a large historical warm bias is likely to overestimate the projected warming. Taking the linear regression as a proof of concept, the intercept at a zero bias in Fig. 5 represents the best multi-model estimate of the warming, and the slope describes a scaling factor of the extra warming that would result from the systematic bias. The increased slope with increased radiative forcing in Fig. 5 demonstrates that the systematic warm bias in the historical simulations will be proportionally amplified with the magnitude of the overall warming. The slope for the three RCPs (0.27, 0.31, and 0.41) corresponds well with the best multi-model estimate of warming (1.20, 2.43, and 5.16 K). The dependence of the slope with temperature is about 4% per degree of warming ($r^2 = 0.999$, $P < 0.001$). This is probably due to the climate feedbacks with warming, such as land–atmosphere interaction, water vapor feedback and cloud feedback[3].

Based on the robust relationships between future warming and the identified present-day warm bias, we are able to make use of this linearity to empirically correct the potential uncertainties in the projected warming (see "Methods" for details). As expected, this procedure leads to decreased warming accompanied by

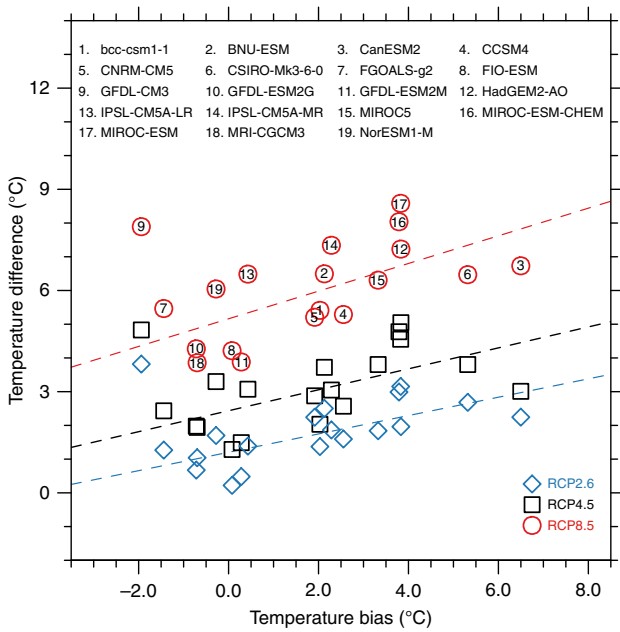

**Fig. 5** Relationship between temperature changes and temperature bias. The relationships between temperature changes (2080–2099 relative to 1981–2000) and temperature bias for the three RCP scenarios. RCP2.6, RCP4.5, and RCP8.5 are represented by blue, black, and red symbols. Dash lines indicate the linear fit to each scenario excluding model 9. Numbers with circle represent the correspondent models. The regression equations for the three RCP scenarios are: $\delta T = 0.27 T_{bias} + 1.20$ ($r^2 = 0.54$; $P < 0.01$), $\delta T = 0.31 T_{bias} + 2.43$ ($r^2 = 0.40$; $P < 0.01$), and $\delta T = 0.41 T_{bias} + 5.16$ ($r^2 = 0.44$; $P < 0.01$), for RCP2.6, RCP4.5, and RCP8.5 respectively

reduced inter-model spread simultaneously (Fig. 6a). The temperature correction results in a mean reduction of warming by 0.7 K and a reduced inter-model spread after the correction, implying a convergence in their warming projections. The overall improved model agreement indicates more confidence in the future projections. In terms of the future projected temperature, which is more relevant for the public, the model systematic bias needs to be considered. Taking RCP8.5 scenario as an example, after both corrections, the future MMM projected temperature will be 28.0 °C instead of 30.5 °C (Fig. 6b). Without the corrections, current GCMs are likely overestimating the future projected temperature over the CUS due to the inherent systematic bias associated with model physics. Overall, the corrections not only substantially narrowed the intermodel differences of the projected future climate, but also altered the quantitative nature of future changes.

**Discussion**

As expected, the situation for precipitation change is more complicated without a distinct dependence on precipitation bias (correlations are not statistically significant for the three RCPs, Supplementary Fig. 4) and warrants further investigation. However, if we assume the correspondence between precipitation bias and temperature bias (as shown in Fig. 2a) still holds in the future, we can adjust precipitation changes based on the temperature correction we proposed. There is no reason to think of this assumption as invalid. After the correction, models projected a slight wetting instead of drying for all three RCP scenarios (Supplementary Fig. 5a). The original MMM projection in RCP8.5 is about 2.5 mm d$^{-1}$, which is substantially drier than the historical observation (3.48 mm d$^{-1}$ averaged over the period

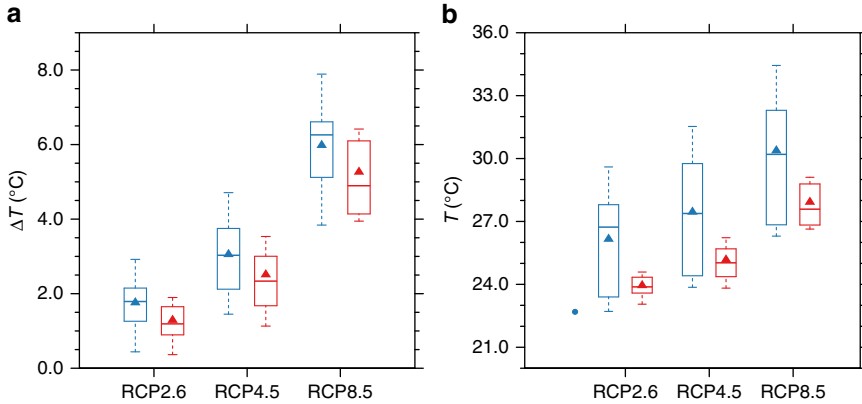

Fig. 6 Bias correction for temperature. **a** Boxplot of temperature changes with (red)/without (blue) bias correction for the three scenarios. **b** As in **a** but for absolute temperature. Solid dots in the first columns represent the observations (averaged over 1980–1999). The boxes indicate 10%-quantile, 25%-quantile, median, 75%-quantile, and 90%-quantile, respectively. Solid triangles stand for the respective multi-model mean

1980–1999). After further considering the systematic dry bias, the adjusted MMM is 3.51 mm d$^{-1}$ (Supplementary Fig. 5b), indicating a negligible increase of future summer precipitation over the CUS. We believe that the adjusted projections are more credible than the default projections considering the large inherent model systematic bias over this region. This can have important implications for appropriate decision makings and adaptation guidance.

To conclude, most climate models suffer from systematic biases. This work aims to better understand the cause of the warm/dry bias over the CUS during summertime, which is highly coherent among the latest GCMs. Our investigation suggests that the cause for the identified warm/dry bias is likely triggered by the precipitation deficit arising from the model's failure in capturing the large precipitation events, followed by strong and complex land–atmosphere interactions on different time scales. Other model deficiencies, such as underestimated shallow cumulus and misrepresented soil processes, may also initiate or reinforce the warm and dry bias, but the deficiency in the large precipitation events in models is at least one of the leading causes of the bias.

## Methods
**Linear fits**. All scatter plots in this study are fitted using linear functions in a least-square sense to assess the dependence of $Y$ on $X$, i.e.,

$$Y = \alpha X + \beta.$$

Where $\alpha$ indicates the linear regression coefficient and $\beta$ indicates the corresponding intercept.

**Cooling effects**. The cooling effect on temperature of individual rainfall event is compared within a week under three different rainfall intensities, i.e., daily rainfall larger than 10 mm, daily rainfall <10 mm but larger than 1 mm, and daily rainfall less than 1 mm but larger than 0.1 mm, respectively. The decrease in temperature is calculated with respect to the day prior to the rainfall event.

**Wet-dry precondition composite**. The wet precondition composite requires five consecutive non-precipitating days following a rainfall event (with daily rainfall ≥1 mm). The dry precondition composite requires five consecutive non-precipitating days following prior five consecutive non-precipitating days. The latter five consecutive non-precipitating days in each composite are used to calculate the composite average. Prior five consecutive non-precipitation days in dry precondition are selected to minimize the influence of a preceding rainfall event in consideration of the soil could impact the land–atmosphere feedback at lags between 4 and 8 days after a rainfall event[11].

**Bias correction**. The correction for the temperature change is straightforward. Once we have the scaling factor, we simply subtract the correction term, i.e., model historical bias times the scaling factor (the slope in Fig. 5), from the model original projected warming for each model. Correction for precipitation here is based on the correspondence of temperature bias and precipitation bias identified in Fig. 2a.

For each model, precipitation correction term is calculated from the linear equation: $\delta(T_{bias}) = -3.28\delta(P_{bias})$, where $\delta$ is the bias correction term for the future scenario. And similarly, these correction terms are added to the model original projected change of precipitation to get the corrected precipitation change.

**Code availability**. All the plots in this study are made using NCAR Command Language (NCL; http://www.ncl.ucar.edu/). The data sets and scripts generated are available from the corresponding author upon request.

**Data availability**. Monthly near-surface air temperature and precipitation data from 19 models contributing to the CMIP5[14] have been used (see Table 1 for details, http://pcmdi9.llnl.gov), including simulations of historical run (1850–2005) and three Representative Concentration Pathway (RCP) scenarios (2006–2099). Among them, 15 models also provide daily precipitation. All data have been linearly interpolated to a native 1°×1° latitude-longitude grid (~100-km spatial resolution), and monthly means are computed over a box centered over the central U.S. (see Fig. 1 for the location of the box). We use the Global Precipitation Climatology Project (GPCP; http://precip.gsfc.nasa.gov/)[29] and the recently compiled Climate Research Unit (CRU; http://www.cru.uea.ac.uk/data) Version 3.23 products[30] during the period of 1979–2005 for the comparison with the historical simulations. The ARM (https://www.arm.gov/sites/sgp)[15] archive at SGP site (36° 36′ 18.0″ N, 97° 29′ 6.0 ″ W) provides vast variables at hourly frequencies. Variables, including cloud fraction, surface downward and upward radiant fluxes, sensible and latent heat fluxes, precipitation, and surface meteorology such as temperature, and relative humidity from ARMBE data sets during 1996–2010 have been used in this study.

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

## Acknowledgements

We gratefully acknowledge the CMIP5 (listed in Supplementary Table 1) for providing public access to these data. We further thank the climate modeling groups for producing the model output and making it available. We also greatly appreciate the US Department of Energy (DOE) ARM Program for supplying the ARMBE data sets. This work was supported by the Ministry of Science and Technology of China (grant 2013CBA01805, 2016YFB02008, 2014CB441303), Major Science and Technology Foundation Program of Ministry of Education, China (Special support for National Supercomputing Center in Guangzhou, Sun Yat-sen University), and by the Biological and Environmental Research Division of the Office of Sciences of the US DOE.

## Author contributions

Y.L. and M.Z. developed the essential research idea. W.D. performed the analysis and wrote the initial draft of the paper. All authors contributed to the interpretation of the results and the preparation of the manuscript.

## Additional information

**Competing interests:** The authors declare no competing financial interests.

**Change history:** A correction to this article has been published and is linked from the HTML version of this paper.

