## [Peer Review File · Nature Communications]

Reviewers' comments:

Reviewer #1 (Remarks to the Author):

This is a very interesting paper, worthy of publication once a few issues have been addressed.

The authors state, quite rightly that:

"Following a heavy precipitation event, temperature drops quickly and substantially and then recovers gradually (Supplementary Fig. 2). The stronger the rainfall, the larger the magnitude of the temperature decrease, and the longer duration of the cooling effect. This suggests that surface temperature following a rain event will be misrepresented if the model could not simulate the precipitation accurately, especially for strong rainfall events. Consequently, the warm bias can stem from a precipitation deficit. Once an initial dry bias is formed, the buildup of the warm bias can occur through positive land-atmosphere feedback processes: A deficit in precipitation not only reduces cooling due to evapotranspiration^{1, 4}, but also reduces cloud formation and soil moisture³, leading to increased solar radiation⁴, thus the amplification of surface warming, and less precipitation."

However, although, the analysis presented, does show that a precipitation deficit leads a temperature rise, I do not feel that the mechanisms for this interaction are sufficiently well discussed. I feel the authors, suggest a little too much that it is simply the direct result of the missing precip leading to reduced evaporative cooling at the surface.

In particular, is it the deficit in precipitation reaching the surface and subsequent reduced cooling from evaporation that is the problem, or is the reduced shallow convective cloud cover on days after a missed precipitation event that leads to too much solar radiation reaching the surface. Similarly, if a MCS precipitation event is missed, there will also presumably be a deficit in the convectively generated cirrus anvil and hence the model will have too much short-wave reaching the surface the next day. It could also be that MCS, and their associated cold-pools, lead to enhanced mixing in the boundary layer and enhanced evaporative cooling after their passage. So yes, I can see how there is a link between missed precip event and too warm temperatures, but I do not feel the authors have provided a convincing explanation of what the physical mechanism is that is actually leading to the problem. i.e. if we suddenly came up with a GCM cumulus parametrization scheme that could propagate systems across the plains, would that fix all the problems. Or would we still have issues because the shallow convective clouds, cirrus anvils, soil properties etc were poorly represented.

The main point of the paper however, I understand to be the identification of the correlation between precip and temperature biases and the ability to bias correct the CMIP ensembles. This part of the paper sounds very worthwhile to me and is certainly worth publishing.

Typographical issues:

page2, line 9 "failure of", not "for"

page2, line 25, should there be a word "model" here, i.e. "using results from current climate MODELS for years to come"

page4, line 25, no need for capital A after colon.

page7, line 1, "It is" not "it's", we are trying to be formal.

Figure 1, caption "seasonal evolution of temperature and precipitation biases FOR"

Supplementary figure 2, I presume that the 7 dots are every day (since the text mentions a week). I understand this is partially a schematic figure, but could the x-axis in the 3 sub-panels be labelled or described in the text.

Reviewer #2 (Remarks to the Author):

Review of the manuscript entitle "Model dry and warm bias over the central U.S.: its cause and impact on the future climate projections" by Yanluan Lin and co-authors

This paper address an important topic, the warm bias over continents, but is based on a great mistake, a confusion in the definition of "warming". When one considers the "warming at the end of the 21st century", one should consider the difference between the temperature at the end of the 21st century ($T(2100)$) and the temperature at another period, for instance at year 2000, $T(2000)$. This is what is done in all the papers that are referenced in the introduction of this manuscript. However, in the rest of this manuscript, the authors consider the "warming" simulated by model as the difference between the temperature simulated by models at the end of the 21st century and the temperature currently observed. Therefore this "warming" includes, by definition, the bias of the temperature simulated by models in 2000 in addition to the the actual warming simulated by models between year 2000 and 2100 (i.e. $T(2100)-T(2000)$). Even if all the models simulated exactly the same warming between year 2000 and 2100 but have different bias in simulating the current temperature, the "warming" as defined by the authors would be different among models. This is almost what shows Fig. 3 that displays the absolute temperature at the end of the century as a function of current bias. And this is even more true for the precipitation displayed on the same figure 3. A slope of 1 (which is almost the case in this manuscript) indicates that the temperature and precipitation have the same difference among models now and at the end of the century. In this study, the main difference originates from the bias in current climate and not from future warming actually simulated by models.

In the following sentence "Under the RCP8.5 scenario, instead of projected severe drying (~30% reduction) and strong warming of 7.8 K at the end of 21st century, models suggest negligible change (~3%) of precipitation and a smaller warming of 5.5 K in summer over the CUS after the correction", the "severe drying" and the "strong warming" is only due to the fact that authors the bias in their estimate. The reduction of the "drying" and the "warming" they show is mostly due to the fact they remove the model bias. And again, nobody, except the authors of this study, include the mean bias when computing the change simulated by models.

Therefore I recommend to reject this paper.

We thank the two reviewers for their careful reading and constructive comments, which have helped to greatly improve the analysis and presentation of the study. Two major comments were raised by the reviewers. One is the need of more discussion on the argument that precipitation deficit leads the subsequent temperature rise. The other is the need of clarification between the warming and the projected temperature. In this revision, we have done substantial work using ARM Best Estimate (ARMBE) archive at SGP to provide in-depth discussion of the relationship between precipitation bias and temperature bias to address the first comment. For the second comment, we improved the bias correction procedure for temperature: a correction of projection induced by the model systematic bias and the systematic bias itself. We also included the projected temperature in the future after the correction instead of just showing the corrected temperature change (i.e., warming projected by the models relative to their historical simulations). For more details, please refer to the detailed responses to each reviewer in the following.

Reviewers' comments:

Reviewer #1 (Remarks to the Author):

This is a very interesting paper, worthy of publication once a few issues have been addressed.

The authors state, quite rightly that:

"Following a heavy precipitation event, temperature drops quickly and substantially and then recovers gradually (Supplementary Fig. 2). The stronger the rainfall, the larger the magnitude of the temperature decrease, and the longer duration of the cooling effect. This suggests that surface temperature following a rain event will be misrepresented if the model could not simulate the precipitation accurately, especially for strong rainfall events. Consequently, the warm bias can stem from a precipitation deficit. Once an initial dry bias is formed, the buildup of the warm bias can occur through positive land-atmosphere feedback processes: A deficit in precipitation not only reduces cooling due to evapotranspiration^{1, 4}, but also reduces cloud formation and soil moisture³, leading to increased solar radiation⁴, thus the amplification of surface warming, and less precipitation."

However, although, the analysis presented, does show that a precipitation deficit leads a temperature rise, I do not feel that the mechanisms for this interaction are sufficiently well discussed. I feel the authors suggest a little too much that it is simply the direct result of the missing precipitation leading to reduced evaporative cooling at the surface.

In particular, is it the deficit in precipitation reaching the surface and subsequent reduced cooling from evaporation that is the problem, or is the reduced shallow convective cloud cover on days after a missed precipitation event that leads to too much solar radiation reaching the surface? Similarly, if a MCS precipitation event is missed, there will also presumably be a deficit in the convectively generated cirrus anvil and hence the model will have too much short-wave reaching the surface the next day. It could also be that MCS, and their associated cold-pools, lead to enhanced mixing in the boundary layer and enhanced evaporative cooling after their passage. So yes, I can see how there is a link between missed precipitation event and too warm temperatures, but I do not feel the authors have provided a convincing explanation of what the physical mechanism is that is actually leading to the problem, i.e. if we suddenly came up with a GCM cumulus parametrization scheme that could propagate systems across the plains, would that fix all the problems. Or would we still have issues because the shallow convective clouds, cirrus anvils, soil properties etc. were poorly represented.

The main point of the paper however, I understand to be the identification of the correlation between precipitation and temperature biases and the ability to bias correct

the CMIP ensembles. This part of the paper sounds very worthwhile to me and is certainly worth publishing.

Response:

We thank the reviewer for his/her constructive suggestions. We agree with the reviewer that the missing or underestimate of propagating precipitation events is not the only cause for the warm bias. Other deficiencies in the model, such as underestimated shallow cumulus, misrepresentation of land soil properties and processes, and anvil clouds, could all contribute to the warm bias. In this revision, using the comprehensive data at ARM SGP, we attempted to quantify the direct effect (evaporative cooling) of precipitation and other processes indirectly associated with rainfall on temperature. For the direct rainfall impacts on temperature, we compare various surface energy budget terms and cloud fractions between the precipitating days and non-precipitating days (Fig. 3, Table 1). In addition, to quantify indirect impacts of rainfall on temperature during the following days, we contrast the surface energy budget terms and cloud fractions between two composites: a wet precondition composite requiring 5-consecutive non-precipitating days followed by a rainfall event and another dry precondition composite requiring 5-consecutive non-precipitating days followed by prior 5-consecutive non-precipitating days. In this way, the two composites enable us to capture and quantify the indirect rainfall impact without overlapping days between the two composites (Fig. 3, Supplementary Fig. 3-4, Table S2, S3).

Accordingly, we have included the following in the text with a new Fig. 3, a new Table 1, Supplementary Fig. 3-4, and Supplementary Table 2-3.

#P5L12-P6L20:

Besides the precipitation bias, other deficiencies reported in the model over this area, such as underestimated shallow cumulus²³, misrepresentation of land soil properties and land-atmosphere processes²⁴, and possibly misrepresented anvil clouds²⁵, could also contribute to the warm bias. Once an initial dry bias is formed, the buildup of the warm bias can occur through a series of land-atmosphere feedback

processes on different time scales: a deficit in precipitation not only alters evaporative cooling due to evapotranspiration^{1, 4}, but also indirectly reduces attendant cloud formation and infiltrated soil moisture^{3, 9, 10, 16}, leading to increased solar radiation⁴, and thus the amplification of surface warming during the following days. The temperature, from an energetic perspective, is predominantly constrained by the radiant and turbulent fluxes. These fluxes are influenced by several environmental factors on different time scales²⁶, including soil moisture, relative humidity, and cloud cover that could be strongly impacted by the preceding nocturnal precipitation. To quantify the impacts of different rainfall associated processes on temperature, we utilize the comprehensive variables from ARM Best Estimate (ARMBE) data archive for detailed analysis. We first compare the diurnal changes in the surface temperature, cloud fraction, and surface energy budget terms between precipitating days and non-precipitating days (Table 1). As shown in the diurnal evolution (Supplementary Fig. 3), the temperature contrast between the different composites peaks during daytime with the value to be 4.8 K (3.4 K if we take the daily average). This indicates that although the reduced temperature in rainy days may partly result from the direct rain evaporative cooling including cold pool associated with the MCS passage, it mainly results from the processes during the daytime. All the daytime radiant and turbulent fluxes are smaller during the precipitating days than those during the non-precipitating days. Clouds play a crucial role in regulating the energy budget given its strong interaction with both solar and infrared radiation. Shallow cumulus, usually occurring during daytime with the growth of the boundary layer and a gradually lifted cloud base under 3 km, has a typical fraction of 10%-12% over this area. It strongly impacts the shortwave radiation. The cloud fraction of shallow cumulus is larger than 15% during precipitating days while it is about 5% during non-precipitating days (Fig. 3a and 3b). The total cloud fraction is about 2 times larger in precipitating days compared to non-precipitating days. This leads directly to a reduction in net shortwave absorption of 107.2 W m^{-2} with 32.4 W m^{-2} smaller net longwave emission. The net reduced radiative heating (74.8 W m^{-2}) is approximately balanced by the decreased latent heat (37.9 W m^{-2}) and sensible heat (36.1 W m^{-2})

fluxes (the residual ground heat flux term is usually small relative to other components). We further divide precipitating days into three categories by intensity (Supplementary Table 2). The average differences of nocturnal precipitation amounts correlate well with the correspondent daytime temperature changes with $r=-0.92$ ($P<0.05$), indicating that the larger the nocturnal precipitation, the larger daytime temperature drop. And consistent with Phillips and Klein¹¹, the net shortwave radiation decreases quasi-linearly with increasing cloud fraction while a highly positive correlation between the net surface longwave fluxes with cloud cover is found. Sensible and latent heat fluxes vary inversely with cloud fraction (Supplementary Table 2). Switching the averaging periods does not change the conclusion. As a result, on a diurnal time scale, the daytime temperature drop in rainy days is predominately associated with the reduced solar radiation.

Table 1. Contrast of precipitation, temperature, cloud fraction, sensible heat, latent heat, and radiation terms between precipitating days and non-precipitating days. The precipitation is averaged between 0000 and 0600 LST while the other variables are averaged between 0600 and 1800 LST. The radiant and turbulent fluxes are positive downward. The numbers in the parentheses indicated the sample size.

	P	T	Total cloud	LH	SH	Net SW	Net LW	
	(mm/d)	(°C)	(%)	(W/m ²)	(W/m ²)	(W/m ²)	(W/m ²)	
Precipitating			9.99 (low)			396.07	405.46	
days	22.08	25.43	12.85 (mid)	62.96	-176.83	-74.57	317.72	-49.95
(288)			39.64 (high)			-78.35	-455.41	
						(downward)	(downward)	
						(upward)	(upward)	
Non-precipitating			6.46 (low)			535.88	406.36	
days	0.00	30.22	7.71 (mid)	29.68	-214.72	-110.69	424.88	-82.31
(945)			15.14 (high)			-111.00	-488.67	
						(downward)	(downward)	
						(upward)	(upward)	
Precipitating			3.53 (low)			-139.81	-0.90	
minus	22.08	-4.79	5.14 (mid)	33.28	37.89	36.12	-107.16	32.36
non-precipitating			24.50 (high)			32.65	33.26	
						(downward)	(downward)	
						(upward)	(upward)	

Supplementary Table 2. Changes in precipitation, temperature, cloud fraction, sensible heat, latent heat, and radiation terms between three different rainfall intensities and non-precipitating days. The precipitation is averaged between 0000 and 0600 LST while the other variables are averaged between 0600 and 1800 LST. The numbers in the parentheses indicated the sample size.

	ΔP (mm/d)	ΔT (K)	Δ Total cloud (%)	ΔLH (W/m ²)	ΔSH (W/m ²)	Net ΔSW (W/m ²)	Net ΔLW (W/m ²)		
Large precipitation (134)	42.42	-5.60	3.03 (low)	35.03	-42.85	-47.99	-160.69	-1.62	36.89
			4.17 (mid)				(downward)	(downward)	
			27.75 (high)				(upward)	(upward)	
Moderate precipitation (154)	4.87	-4.09	3.99 (low)	31.63	-33.68	-25.95	-121.06	-0.25	28.30
			6.05 (mid)				(downward)	(downward)	
			21.47 (high)				(upward)	(upward)	
Light precipitation (124)	0.72	-2.73	2.67 (low)	13.29	-2.60	-16.75	-54.99	-3.94	15.69
			3.50 (mid)				(downward)	(downward)	
			7.08 (high)				(upward)	(upward)	

Figure 3 | Composite diurnal variations of cloud fraction. Diurnal cycle of cloud fraction averaged over (a) precipitating days, (b) non-precipitating days, (c) 5-consecutive non-precipitating days under wet precondition, and (d) 5-consecutive non-precipitating days under dry precondition. The sample sizes are 288, 945, 49 and 73, respectively. Contours indicate 5%, 10%, and 15% cloud fraction.

#P6L21-P7L22:

During the following days after a rainfall event, however, the inherent complexities in land-atmosphere interactions might make these processes more complicated. For instance, soil moisture is proved to have profound impacts on land-atmosphere interaction due to its long memory⁸. It could impact cloud formation in the aftermath of a precipitation event even without any rainfall in the following several days. Gradual drying of the soil after a rainfall event has been found to impact the land-atmosphere feedback at lags between 4 and 8 days¹¹. To quantify impacts of rainfall on temperature in the following days, we contrast the surface energy terms between two composites: a wet precondition composite requiring 5-consecutive non-precipitating days following a rainfall event and a dry precondition composite requiring 5-consecutive non-precipitating days following prior 5-consecutive

non-precipitating days. The prior 5-consecutive non-precipitating days in dry precondition composite are adopted here to minimize the influence of preceding rainfall events (see Methods). This classification makes the two composites independent of each other and thus enables us to quantify the rainfall impact on temperature. Compared with dry precondition, surface temperature after the rainfall is about 2.8 K lower under wet precondition, which, from the surface energy budget, is primarily associated with the enhanced latent heat fluxes (Supplementary Table 3). Larger cloud fraction is noted under wet precondition, particularly, shallow cumulus is more prevalent due to the strong coupling with soil moisture (Fig. 3c). In contrast, shallow cumulus is rather limited under dry precondition (Fig. 3d). This indicates that soil properties and land surface processes after the rainfall infiltration can indirectly regulate the subsequent low cloud formation. These features are also evident in the evolution of other variables in the surface energy budget (Supplementary Fig. 4). Low clouds gradually decrease to a level ($\sim 5\%$) similar to that in the dry composite three days after a rainfall event. This is likely due to the subsequent gradual evaporative drying of the soil in the absence of additional precipitation. One thing worth noting is that despite the larger cloud fraction under wet precondition, there is more downward solar radiation near the surface (Supplementary Table 3). This lies in the seasonal variation of the solar insolation at the top of the atmosphere. It decreases progressively from June to September. And wet precondition composite has larger proportion in June (28.6%) than that in dry precondition composite (6.8%). In reality, the scaled surface downward shortwave radiation increases gradually to that under dry precondition, similar to the evolution of cloud fraction (Supplementary Fig. 4e). These results imply that except the immediate temperature drop on rainy days, such an impact could last for several days in the aftermath of a precipitation event due to the land-atmosphere interaction mediated by soil moisture. Other properties, like the lower atmosphere's aerodynamic resistance and humidity saturation deficit, could complicate the strength of land-atmosphere coupling^{11, 27}.

Supplementary Figure 3 | Composite surface temperature diurnal variations. (a) Diurnal cycle of surface temperature averaged over precipitating days and non-precipitating days. (b) Diurnal cycle of surface temperature averaged over non-precipitating days with wet precondition and non-precipitating days with dry precondition. The sample sizes in **a** are 288 and 945 while they are 49 and 73 in **b**. Shading indicates one s.e.m. for each composite. The green areas below are their respective differences.

Supplementary Figure 4. Evolution of (a) sensible heat flux, (b) latent heat flux, (c) low cloud, (d) total cloud, (e) scaled downward shortwave radiation, and (f) net longwave radiation under wet (blue) and dry (red) preconditions. Shading indicates one s.e.m. for each composite. Sample sizes for wet and dry preconditions are 49 and 73, respectively.

Supplementary Table 3. Contrast of solar insolation at the top of atmosphere, surface temperature, cloud fraction, sensible heat, latent heat, and radiation terms between wet and dry preconditions. All the variables are averaged between 0600 and 1800 LST on the following 5-consecutive non-precipitating days. The radiant and turbulent fluxes are positive downward. The numbers in the parentheses indicated the sample size.

	SWDN _{TOA} (W/m ²)	T (°C)	Total cloud (%)	LH (W/m ²)	SH (W/m ²)	Net SW (W/m ²)	Net LW (W/m ²)	
Wet precondition (49)	847.23	30.10	5.25 (low)	26.54	-223.49	-108.46	554.25 (downward)	402.43 (downward)
			6.83 (mid)				-113.43 (upward)	-489.43 (upward)
			14.11 (high)					
Dry precondition (73)	828.53	31.73	4.68 (low)	23.79	-207.00	-115.80	544.12 (downward)	411.33 (downward)
			5.50 (mid)				-116.12 (upward)	-499.47 (upward)
			13.29 (high)					
Wet precondition minus dry precondition	18.70	-1.63	0.57 (low)	2.75	-16.49	7.34	10.13 (downward)	-8.90 (downward)
			1.33 (mid)				-2.69 (upward)	-10.04 (upward)
			0.82 (high)					

Based on the new analysis we did above, we agree with the reviewer that “So even if we suddenly come up with a GCM cumulus parameterization scheme that could propagate systems across the plains, it would not fix all the problems. We still have issues with the formation of shallow cumulus, convective anvils, and soil properties, among others. ” In this sense, efforts for improvements of dry/warm bias over the CUS in next generation of GCMs not only need to be paid in convective parameterization to better simulate the intensity and phase of precipitation, but also need to be paid in these land processes and land-atmosphere interactions. We have now stated in the revised manuscript that other model deficiencies, such as underestimated shallow cumulus and misrepresented soil processes, may also initiate or reinforce the warm and dry

bias, but the deficiency in the large precipitation events in models is at least one of the leading causes of the bias.

Typographical issues:

page2, line 9 "failure of", not "for"

Corrected

page2, line 25, should there be a word "model" here, i.e. "using results from current climate MODLES for years to come"

Corrected

page4, line 25, no need for capital A after colon.

Corrected

page7, line 1, "It is" not "it's", we are trying to be formal.

Corrected

Figure 1, caption "seasonal evolution of temperature and precipitation biases FOR"

Corrected

Supplementary figure 2, I presume that the 7 dots are every day (since the text mentions a week). I understand this is partially a schematic figure, but could the x-axis in the 3 sub-panels be labelled or described in the text.

Response:

We are sorry about the confusion induced by Supplementary Fig. 2. The descriptions are added at #P22L2 for clarity. This is not a schematic figure; instead, they are averaged over three different categories, i.e. large precipitation event (with daily rainfall larger than 10 mm), and moderate rainfall event (larger than 1 but less than 10 mm) and light rainfall event (larger than 0.1 but less than 1 mm). The mean precipitation amounts for the three categories are 27.4, 3.89, and 0.4 mm day⁻¹ and the correspondent sample sizes are 134, 154, and 124, respectively. The updated Supplementary Fig. 2 is shown below:

Supplementary Figure 2. Cooling effects. Temperature change (K, left y-axis) induced by three different intensities of rainfall events (mm day^{-1} , right y-axis) within a week. The temperature changes are calculated with respect to the day prior to the rainfall event. Numbers N indicate the sample sizes in each rainfall categories.

Reviewer #2 (Remarks to the Author):

Review of the manuscript entitle “Model dry and warm bias over the central U.S.: its cause and impact on the future climate projections” by Yanluan Lin and co-authors

This paper address an important topic, the warm bias over continents, but is based on a great mistake, a confusion in the definition of “warming”. When one considers the “warming at the end of the 21st century”, one should consider the difference between the temperature at the end of the 21st century ($T(2100)$) and the temperature at another period, for instance at year 2000, $T(2000)$. This is what is done in all the papers that are referenced in the introduction of this manuscript. However, in the rest of this manuscript, the authors consider the “warming” simulated by model as the difference between the temperature simulated by models at the end of the 21st century and the temperature currently observed. Therefore this “warming” includes, by definition, the bias of the temperature simulated by models in 2000 in addition to the the actual warming simulated by models between year 2000 and 2100 (i.e. $T(2100)-T(2000)$). Even if all the models simulated exactly the same warming between year 2000 and 2100 but have different bias in simulating the current temperature, the “warming” as defined by the authors would be different among models. This is almost what shows Fig. 3 that displays the absolute temperature at the end of the century as a function of current bias. And this is even more true for the precipitation displayed on the same figure 3. A slope of 1 (which is almost the case in this manuscript) indicates that the temperature and precipitation have the same difference among models now and at the end of the century. In this study, the main difference originates from the bias in current climate and not from future warming actually simulated by models.

In the following sentence “Under the RCP8.5 scenario, instead of projected severe drying (~30% reduction) and strong warming of 7.8 K at the end of 21st century, models suggest negligible change (~3%) of precipitation and a smaller warming of 5.5 K in summer over the CUS after the correction”, the “severe drying” and the “strong warming” is only due to the fact that authors the bias in their estimate. The reduction of the ”drying” and the “warming” they show is mostly due to the fact they remove the model bias. And again, nobody, except the authors of this study, include the mean bias when computing the change simulated by models.

Therefore I recommend to reject this paper.

Response:

We thank the reviewer for pointing out this “great mistake”. We agree with the reviewer that warming is conventionally defined as the difference between the

warming simulation and the historical simulation, i.e., $T_{\text{warm}}-T_{\text{hist}}$. Although the correction we did before actually included two parts: one is the systematic bias and the other is the amplification of the systematic bias with warming. We did not make it clear in the text and the plots, especially in Fig. 3, which directly shows the relation between T_{bias} and $T_{\text{projection}}$. We are sorry about this confusion. In the revision, we have explicitly divided the T correction into two parts: one is the systematic bias, and the other is the amplification of the systematic bias associated with the warming (new Fig. 4).

In this revision, we follow the reviewer's suggestion and do the correction based on the relationship between the warming itself ($T_{\text{warm}}-T_{\text{hist}}$) and T_{bias} (Fig. 4a). We can see the warming depends on the bias with larger slopes for RCP8.5 than RCP2.6 and RCP4.5. This indicates the T_{bias} will be amplified in a warmer climate. This bias-induced amplification should be corrected for the projected warming (Fig. 4b). Finally, we presented the adjusted temperature projection after considering the systematic temperature bias in Fig. 4c. No distinct dependence of projected precipitation change on precipitation bias was found (new Supplementary Fig. 5a) since precipitation change is more complicated and is not necessarily dominated by the bias itself. However, assuming that the close correspondence between precipitation bias and temperature bias still holds in the future, we attempted to do a precipitation adjustment based on the temperature correction we did. This part has been included in the Discussion section with new Supplementary Fig. 5.

Accordingly, we have included the following in the text with a new Fig. 4, a new Supplementary Table 4, and Supplementary Fig. 5.

#P8L16-P9L14:

To estimate the impact of model systematic warm bias on future projections, we regress the projected temperature change with the identified systematic biases. We note robust linear relationships between temperature bias and its future projected change in all three RCP scenarios (Fig. 4a), with linear regression coefficient to be 0.27,

0.31, 0.41 and $r^2=0.54, 0.40, \text{ and } 0.44$ ($P<0.01$). The linear least-square fitting lines exclude model 9, which has known limitations for too large climate sensitivity²⁸ (but is included here for completeness). This suggests that a model with a large historical warm bias is likely to overestimate the projected warming. Taking the linear regression as a proof of concept, the intercept at a zero bias in Fig. 4a represents the best multi-model estimate of the warming, and the slope describes a scaling factor of the extra warming that would result from the systematic bias. The increased slope with increased radiative forcing in Fig. 4a demonstrates that the systematic warm bias in the historical simulations will be proportionally amplified with the magnitude of the overall warming. The slope for the three RCPs (0.27, 0.31, and 0.41) corresponds well with the best multi-model estimate of warming (1.20, 2.43, and 5.16 K). The dependence of the slope with temperature is about 4% per degree of warming ($r^2=0.999, P<0.001$). This is probably due to the climate feedbacks with warming, such as land-atmosphere interaction, water vapor feedback and cloud feedback³.

Based on the robust relationships between future warming and the identified present-day warm bias, we are able to make use of this linearity to empirically correct the potential uncertainties in the projected warming (see Methods for details). As expected, this procedure leads to decreased warming accompanied by reduced inter-model spread simultaneously (Fig. 4b). The temperature correction results in a mean reduction of warming by 0.7 K and a reduced inter-model spread after the correction, implying a convergence in their warming projections. The overall improved model agreement indicates more confidence in the future projections. In terms of the future projected temperature, which is more relevant for the public, the model systematic bias needs to be considered. Taking RCP8.5 scenario as an example, after both corrections, the future MMM projected temperature will be 28.0 °C instead of 30.5 °C. Without the corrections, current GCMs are likely overestimating the future projected temperature over the CUS due to the inherent systematic bias associated with model physics. Overall, the corrections not only substantially narrowed the intermodal

differences of the projected future climate, but also altered the quantitative nature of future changes.

Figure 4 | Bias correction for temperature. (a) The relationships between temperature changes (2080-2099 relative to 1981-2000) and temperature bias for the three scenarios. RCP2.6, RCP4.5, and RCP8.5 are represented by blue, green, and red symbols. Dash lines indicate the linear fit to each scenario excluding model 9. The regression equations are listed at the bottom of the plot. Numbers with circle represent the correspondent models. (b) Boxplot of temperature changes with (red)/without (blue) bias correction for the three scenarios. (c) As in (b) but for absolute temperature. Solid dots in the first columns represent the observations (averaged over 1980-1999). The boxes indicate 10%-quantile, 25%-quantile, median, 75%-quantile, and 90%-quantile, respectively. Solid triangles stand for the respective MMM.

#P9L17-P9L32:

As expected, the situation for precipitation change is more complicated without a distinct dependence on precipitation bias (correlations are not statistically significant for the three RCPs, Supplementary Fig. 5a) and warrants further investigation. However, if we assume the correspondence between precipitation bias and temperature bias (as shown in Fig. 1a) still holds in the future, we can adjust precipitation changes based on the temperature correction we proposed. There is no reason to think of this assumption as invalid. After the correction, models projected a slight wetting instead of drying for all three RCP scenarios (Supplementary Fig. 5b). The original MMM projection in RCP8.5 is about 2.5 mm day⁻¹, which is substantially drier than the historical observation (3.5 mm day⁻¹ averaged over the period 1980-1999). After further considering the systematic dry bias, the adjusted MMM is 3.7 mm day⁻¹ (Supplementary Fig. 5c), indicating a slight increase of future

summer precipitation over the CUS. We believe that the adjusted projections are more credible than the default projections considering the large inherent model systematic bias over this region. This can have important implications for appropriate decision makings and adaptation guidance.

Supplementary Figure 5. Bias correction for precipitation. (a) The relationships between precipitation changes (2080-2099 relative to 1981-2000) and precipitation bias for the three scenarios. RCP2.6, RCP4.5, and RCP8.5 are represented by blue, green, and red symbols. Dash lines indicate the linear fit to each scenario. The regression equations are listed at the bottom of the plot. Numbers with circle represent the correspondent models. (b) Boxplot of precipitation changes with (red)/without (blue) bias correction for the three scenarios. The bias correction is based on the correspondence between precipitation bias and temperature bias shown in Fig. 1a. (c) As in b but for absolute precipitation. Solid dots in the first columns represent the observations (averaged over 1980-1999). The boxes indicate 10%-quantile, 25%-quantile, median, 75%-quantile, and 90%-quantile, respectively. Solid triangles stand for the respective MMM.

Reviewers' comments:

Reviewer #1 (Remarks to the Author):

Thank your for addressing the issues I raised. I believe the paper is now suitable for publication.

Reviewer #2 (Remarks to the Author):

Review of the revised version of the manuscript entitle "Model dry and warm bias over the central U.S.: its cause and impact on the future climate projections" by Yanluan Lin and co-authors

The authors significantly improved the revised version of their manuscript for which my main concern was the mixing of the bias (relative to current climate) and the spread of future projections, for both temperature and precipitation. Now the text is much more clear and I appreciate the improvements made by the authors. The results and much less impressive but are still significant and consistent with what can be deduced from other studies.

A strength of the paper is the detailed analysis of the mechanism at the origin of the warm bias. The authors clearly demonstrate that models with dry bias are too warm. They also perform a very nice analysis of the impact of rainfall using ARM Best Estimate data.

I still have two questions:

1) The authors claim that the warm bias is not only due to the lack of rain but to the lack of heavy rain. They identify a cooling after rainfall events using observations, and the amplitude of the cooling is larger for large rainfall events (Figure S-2). However, the warming after this temporary cooling is always present and large, even if the rainfall is small and the corresponding cooling is almost non-existent. What is the reason of this warming? Is it an effect of clouds that is discuss later in the paper but not in this context? Is it an artefact of the method?

2) Corrections for projections. The corrections for projected temperature are now convincing, but I still have questions for precipitations. The method to correct precipitations is not presented with enough details in the text and the results are difficult to interpret. In Fig. S-5, precipitations change is positive for all scenarios: the increase is the largest for RCP2.6 (small warming) and smaller for RCP8.5 (large warming). In other words, results show an increase of precipitations for a small climate change, and then this increase reduces when climate change becomes larger. Is this a real effect (in don't believe it, but if it is the case this would be very interesting and need to be better highlight and interpret) or an artefact of the correction methods? Using the same method, does precipitations simulated by models correspond to observation?

Minor comments:

line 9-10: please specify that this reduction of 2.5K is mainly due to the removal of the bias.

Line 35 : fraction, not faction

Reviewer #2 (Remarks to the Author):

Review of the revised version of the manuscript entitled “Model dry and warm bias over the central U.S.: its cause and impact on the future climate projections” by Yanluan Lin and co-authors

The authors significantly improved the revised version of their manuscript for which my main concern was the mixing of the bias (relative to current climate) and the spread of future projections, for both temperature and precipitation. Now the text is much more clear and I appreciate the improvements made by the authors. The results are much less impressive but are still significant and consistent with what can be deduced from other studies.

A strength of the paper is the detailed analysis of the mechanism at the origin of the warm bias. The authors clearly demonstrate that models with dry bias are too warm. They also perform a very nice analysis of the impact of rainfall using ARM Best Estimate data.

I still have two questions:

- 1) The authors claim that the warm bias is not only due to the lack of rain but to the lack of heavy rain. They identify a cooling after rainfall events using observations, and the amplitude of the cooling is larger for large rainfall events (Figure S-2). However, the warming after this temporary cooling is always present and large, even if the rainfall is small and the corresponding cooling is almost non-existent. What is the reason of this warming? Is it an effect of clouds that is discussed later in the paper but not in this context? Is it an artefact of the method?

Response:

The warming after the temporary cooling is because without the precipitation events and the associated increase of cooling due to surface evaporation and cloud shielding of solar radiation, there is an increase of net downward radiant flux after the rainfall events. We have added the new Supplementary Fig. 2b to show this (see below). After a precipitation event, the net radiant heat flux increases quickly. Though temperature is not fully constrained by the net radiant heat flux, the change of turbulent heat flux after the rainfall events, by contrast, is relatively smaller. This can also be seen in Table 1 and Supplementary Table 2. Therefore, the direct cause of the recovery of surface temperature is more by clouds and solar radiation than by evaporation after the precipitation events, even though evaporation can indirectly impact clouds and then radiation. Their roles in temperature variation are further investigated by carefully exploring the land-atmosphere feedback processes on different time scales in the following section. In this context, we have revised the text by adding this discussion from your valuable comment (P4L32-34). Another reason for the quick recovery of temperature, especially for the small rainfall category, is partly due to the composite method itself. For this composite, we have no

requirement for the rainfall before the event. In fact, it was found that one third of the events in light rainfall category ($0.1 \leq P < 1 \text{ mm day}^{-1}$) generally occurred after a larger rainfall event ($P \geq 1 \text{ mm day}^{-1}$). As a result, there is a quick recovery of temperature for this category since there is a temperature drop before the event. This caveat has been mentioned in the text and addressed in details using a more restricted composite (see Supplementary Fig. 4 and Supplementary Table 3).

Supplementary Figure 2. Cooling effects. (a) Relative change of temperature (K, left y-axis) and (b) relative change of net downward radiant heat flux (W m^{-2} , left y-axis) after three different intensities of rainfall events (mm day^{-1} , right y-axis) within a week. The temperature and net downward radiant heat flux changes are calculated with respect to the day prior to the rainfall event. Numbers N indicate the sample sizes in each rainfall categories.

2) Corrections for projections. The corrections for projected temperature are now convincing, but I still have questions for precipitations. The method to correct precipitations is not presented with enough details in the text and the results are difficult to interpret. In Fig. S-5, precipitations change is positive for all scenarios: the increase is the largest for RCP2.6 (small warming) and smaller for RCP8.5 (large warming). In other words, results show an increase of precipitations for a small climate change, and then this increase reduces when climate change becomes larger. Is this a real effect (in don't believe it, but if it is the case this would be very interesting and need to be better highlight and interpret) or an

artefact of the correction methods? Using the same method, does precipitations simulated by models correspond to observation?

Response:

The current correction for precipitation is based on the assumption that the correspondence between precipitation bias and temperature bias (as shown in Fig. 1a) holds in the future. The equation $T_{\text{bias}} = -3.28P_{\text{bias}} - 0.66$ was used for precipitation correction in the previous version. When we apply it to historical simulation, the corrected mean value is indeed much more close to the observation though uncertainty still exists in model spread (see Figure R1 below). But for the future climate change, after considering your comments carefully, we think the scaling relationship $\delta(T_{\text{bias}}) = -3.28\delta(P_{\text{bias}})$, where δ is the *bias correction*, for the future scenarios is more appropriate. The intercept value (-0.66) derived from the historical simulation implies the existence of additional factors that may contribute to the warm bias. It conforms to the current climate, but may change in a warming scenario (at least with less confidence). Thus we revised the method to use the scaling correspondence equation $\delta(T_{\text{bias}}) = -3.28\delta(P_{\text{bias}})$ to correct the precipitation projection (see Supplementary Fig. 5 and the revised Method). In other words, we don't include the intercept (-0.66) in the new correction for precipitation. As a result, this would induce a reduction of $\sim 0.2 \text{ mm day}^{-1}$ correction compared to the previous correction method. After correction, the projected precipitation change by the end of the 21st century in this region is statistically similar in all three scenarios and nearly neutral, in contrast to the drying projection in RCP8.5 without the bias correction. The reason for the diminished difference of precipitation changes for the three scenarios is because the correction is relatively larger for RCP8.5 because of the larger amplification of temperature bias and thus the larger precipitation correction for RCP8.5. The neutral precipitation change after the correction is consistent with Supplementary Fig. 5a, in which the intercepts of all three regression lines are within the range of $[-0.1, 0.1]$ at a zero precipitation bias (This implies that the future precipitation change will be rather small if precipitation bias is close to zero). This further confirms the robustness of the correction. We thank you for your valuable insights that prompted us to refine the method.

Accordingly, we revised the last sentence in the abstract to be “Instead of a sharp decrease, the projected precipitation by the end of the 21st century is nearly neutral for all scenarios after the correction.” and the adjusted MMM precipitation projection to be 3.51 mm day^{-1} (P9L32).

Figure R1. Bias correction for precipitation in historical simulation. Boxplot of precipitation changes with (red)/without (blue) bias correction for historical simulations. The bias correction is based on the correspondence between precipitation bias and temperature bias shown in Fig. 1a. Solid dots in the first columns represent the observations (averaged over 1980-1999). The boxes indicate 10%-quantile, 25%-quantile, median, 75%-quantile, and 90%-quantile, respectively. Solid triangles stand for the respective MMM.

Supplementary Figure 5. Bias correction for precipitation. (a) The relationships between precipitation changes (2080-2099 relative to 1981-2000) and precipitation bias for the three scenarios. RCP2.6, RCP4.5, and RCP8.5 are represented by blue, green, and red symbols. Dash lines indicate the linear fit to each scenario. The regression equations are listed at the bottom of the plot. Numbers with circle represent the correspondent models. (b) Boxplot of precipitation changes with (red)/without (blue) bias correction for the three scenarios. The bias correction is based on the correspondence between precipitation bias and temperature bias shown in Fig. 1a. (c) As in (b) but for absolute precipitation. Solid dots in the first columns represent the observations (averaged over 1980-1999). The boxes indicate 10%-quantile, 25%-quantile, median, 75%-quantile, and 90%-quantile, respectively.

Solid triangles stand for the respective MMM.

Minor comments:

line 9-10: please specify that this reduction of 2.5K is mainly due to the removal of the bias.

Corrected. The description “mainly resulted from the removal of the warm bias” is added at P2L10 for clarity.

Line 35: fraction, not faction

Corrected.

REVIEWERS' COMMENTS:

Reviewer #2 (Remarks to the Author):

I thank the authors for addressing the issues I raised. I believe the paper is now suitable for publication. However, the quality of the figure is bad and need to be improved.

One spelling error: p2, line 34, intermodal need to be changed in intermodel